# Estimated Costs of the Ipilimumab–Nivolumab Therapy and Related Adverse Events in Metastatic Melanoma

**DOI:** 10.3390/cancers15010031

**Published:** 2022-12-21

**Authors:** Bianca Gautron Moura, Camille L. Gerard, Nathalie Testart, Marian Caikovski, Alexandre Wicky, Veronica Aedo-Lopez, Grégoire Berthod, Krisztian Homicsko, John O. Prior, Clarisse Dromain, Lana E. Kandalaft, Michel A. Cuendet, Olivier Michielin

**Affiliations:** 1Service of Oncology, Cantonal Hospital Fribourg (HFR), Chemin des Pensionnats 1-6, 1700 Fribourg, Switzerland; 2Department of Oncology, Lausanne University Hospital and Agora Translational Cancer Research Center, Rue du Bugnon, 1011 Lausanne, Switzerland; 3Department of Radiology, Lausanne University Hospital (CHUV), Rue du Bugnon 46, 1011 Lausanne, Switzerland; 4Department of Oncology, Monash Medical Centre, 823-865 Centre Road, East Bentleigh, Melbourne, VIC 3165, Australia; 5Service of Oncology, Valais Hospital (CHVR), Avenue Grand Champsec 80, 1951 Sion, Switzerland; 6Department of Physiology and Biophysics, Weill Cornell Medicine, 1300 York Av., New York, NY 10065, USA; 7Swiss Institute of Bioinformatics, UNIL Sorge, 1015 Lausanne, Switzerland

**Keywords:** ipilimumab, nivolumab, costs, immune-related adverse events, immune checkpoint inhibitors, toxicity, immunotherapy, melanoma

## Abstract

**Simple Summary:**

In stage IV melanoma, the combination of ipilimumab and nivolumab improves outcomes compared to single agents. However, it bears an important financial impact on the healthcare system. In this manuscript, we analyze the costs related to the ipilimumab–nivolumab combination, focusing on the immune-related adverse events (irAE) in a real-world setting. Surprisingly, we found that the toxicity cost of ipilimumab and nivolumab are insignificant compared to the total cost of the treatment. Additionally, patients with a Grade 3–4 toxicity have lower costs and better outcomes than patients with Grade 1–2 irAEs. Finally, medication and disease-related hospitalization costs are the major costs of the treatment. These findings are important information for clinicians and healthcare organizations for treatment decisions in immunotherapy.

**Abstract:**

Combined ipilimumab and nivolumab significantly improve outcomes in metastatic melanoma patients but bear an important financial impact on the healthcare system. Here, we analyze the treatment costs, focusing on irAE. We conducted a retrospective analysis of 62 melanoma patients treated with ipilimumab–nivolumab at the Lausanne University Hospital between 1 June 2016 and 31 August 2019. The frequency of irAEs and outcomes were evaluated. All melanoma-specific costs were analyzed from the first ipilimumab–nivolumab dose until the therapy given subsequently or death. A total of 54/62 (87%) patients presented at least one irAE, and 31/62 (50%) presented a grade 3–4 irAE. The majority of patients who had a complete response 12/14 (86%) and 21/28 (75%) of overall responders presented a grade 3–4 toxicity, and there were no responses in patients without toxicity. Toxicity costs represented only 3% of the total expenses per patient. The most significant contributions were medication costs (44%) and disease costs (39%), mainly disease-related hospitalization costs, not toxicity-related. Patients with a complete response had the lowest global median cost per week of follow up (EUR 2425) and patients who had progressive disease (PD), the highest one (EUR 8325). Except for one patient who had a Grade 5 toxicity (EUR 6043/week), we observe that less severe toxicity grades (EUR 9383/week for Grade 1), or even the absence of toxicity (EUR 9922/week), are associated with higher median costs per week (vs. EUR 3266/week for Grade 4 and EUR 2850/week for Grade 3). The cost of toxicities was unexpectedly low compared to the total costs, especially medication costs. Patients with higher toxicity grades had better outcomes and lower total costs due to treatment discontinuation.

## 1. Introduction

The treatment of advanced melanoma was revolutionized over the past decade, as several innovative treatment options that markedly improve responses and/or survival were approved. These include anti-CTLA-4 and anti-PD1 inhibitors, as well as targeted therapies for BRAF-mutant melanoma patients [1,2,3,4,5,6,7,8]. Combined ipilimumab–nivolumab therapy in metastatic melanoma patients can improve response rates (RR) and progression-free survival (PFS) and achieve significant long-term survival benefit [9,10,11]. While the combination has now clearly set a new standard alongside PD-1 single agents, immune-related adverse events (irAEs) are more frequent and exceed what had previously been observed with single immunotherapy agents. As previously reported by Larkin et al., the frequency of Grade 3 or 4 irAEs was significantly higher in the combination arm (59%) compared to the monotherapy arms (23% in the nivolumab group and 28% in ipilimumab group) [9].

While rapid diagnosis and appropriate treatment can resolve most irAEs, severe cases can lead to treatment discontinuation, hospitalization, and even death. Many irAEs can be complex to diagnose and require additional testing, such as biopsies, to determine the exact etiology. In addition, irAEs can appear long after treatment initiation and/or can have long-lasting effects, often underreported in clinical trials [12]. Algorithms have been developed to assist healthcare practitioners to recognize and appropriately manage irAEs [13,14], illustrating the complexity of the task.

The ipilimumab–nivolumab combination has an increased medication cost compared to the monotherapy. Furthermore, the increased toxicity, as attested by the higher incidence of Grade 3–4 irAEs [9], can also be expected to lead to significantly higher indirect costs, including more intensive outpatient follow-ups, more frequent blood tests, outpatient management of irAEs, or even hospitalizations [13,14,15]. On the other hand, some patients achieve a complete response after early treatment discontinuation because of severe irAEs in which case the final cost of the treatment can prove to be even lower than other standard options [16]. As checkpoint inhibitors are poised to have a significant financial impact on the healthcare system, we performed this study to assess the real-world costs of the treatment. We analyzed various cost sources of ipilimumab–nivolumab treatments and focused on the associated irAEs and the costs of managing these in a real-world setting.

## 2. Materials and Methods

### 2.1. Patient Selection and Data Acquisition

The study was approved by the Research Ethics Committee-Vaud Canton, Switzerland (CHUV_DO_CTE_TRP_0005_2017) [17,18,19]. We conducted a single-center, retrospective chart review of 62 patients with a diagnosis of metastatic melanoma that received ipilimumab–nivolumab treatment in Lausanne University Hospital (CHUV), Switzerland. No patients were excluded from the analysis. The patients initiated ipilimumab–nivolumab between the approval of the combination in Switzerland (1 June 2016) and 28 February 2019. The standard regimen given was nivolumab (1 mg/kg) plus ipilimumab (3 mg/kg) every 3 weeks for four doses, followed by nivolumab (3 mg/kg) every 2 weeks for maximum 2 years. Patients were tracked between the first dose of ipilimumab–nivolumab and the therapy given subsequently, death or the end of the follow-up period. The minimum follow-up period was 6 months, and the final evaluations were completed on 31 August 2019. We included only patients with minimal data available (diagnosis date, melanoma type, treatment lines and dates), and who received at least one cycle of ipilimumab–nivolumab at CHUV. If a patient had to be hospitalized in another hospital or had the following treatments in another institution, information was transmitted to CHUV as part of a patient medical follow-up.

Clinical and health-related personal data (age, sex, diagnosis, disease status including staging, presence of brain metastases, radiologic and laboratory results, comorbidities, oncological treatments, including systemic treatments and radiotherapy, medications, and irAEs, as well as mutations and PDL-1 status) were retrieved from patient records by the authorized qualified personnel of the CHUV Oncology Department. Financial data regarding ipilimumab–nivolumab immunotherapy costs were obtained through the CHUV Billing Department. Costs of subsequent treatments were not included in this analysis. Date of death was retrieved through the Swiss Federal Registry for the Persons.

Patients included in the study fulfill the following criteria: male and female; over 18 years old; diagnosed with a melanoma: cutaneous, mucosal, conjunctival, uveal or melanoma of unknown primary (MUP); Stage IV disease; received at least one cycle of the ipilimumab–nivolumab combination; may have received adjuvant treatment prior to the curative treatment; may have received more than 3 lines of treatment; may have been included in clinical trials but the administrated treatment is known (unblinded process).

Because of the high frequency of irAEs reported, surveillance visits with laboratory tests were also performed weekly during the concomitant phase. Between 2016 and 2017 (due to the beginning of treatment implementation in clinical practice), patients also received a phone call from a specialized nurse to check their symptoms. During the monotherapy phase, patients were assessed only before treatments and if judged necessary by the doctors.

Tumor response was assessed every 3 months by chest-abdominal-pelvic CT scan or PET/CT and brain MRI or brain CT scan every 2 to 6 months, depending on the presence of brain metastasis. Subsequently, a retrospective analysis, according to RECIST 1.1 [20], iRECIST [21] and PERCIST, was performed. IrAE description follows the Common Terminology Criteria for Adverse Events (CTCAE) version 4.0. A brain MRI or brain CT scan performed maximum 3 days after ipilimumab–nivolumab start was considered as a baseline exam. Patients had their mutational status defined by pyrosequencing and/or next generation sequencing (NGS) analysis. Tumors that had a BRAF V600E/K mutation were considered NRAS wild-type since these mutations are statistically mutually exclusive.

### 2.2. Evaluation of Adverse Events Related to Immunotherapy and Their Costs

The main irAEs were analyzed at all grades according to the following categories: skin (rash, pruritus and vitiligo), gastrointestinal (diarrhea, gastritis, hepatitis and pancreatitis), endocrine (thyroiditis and hypophysitis), pulmonary (pneumonitis and bronchiolitis), renal (nephritis), myositis, allergies, rheumatologic (Sjögren and arthralgia), cardiac, ocular, and neurological toxicities. We evaluated the irAEs frequency, duration, resource utilization (including multidisciplinary expert opinions, visits with other consultants, need and duration of steroids or 2nd line ISDs, including anti-TNF alpha and mycophenolate mofetil), and frequency of hospitalizations due to toxicity.

Contributions from all interventions were evaluated to measure the estimated cost, namely: oncological medical and nurse visits (i.e., weekly and bi-weekly); laboratory and procedures costs (i.e., blood tests and imaging). Treatment subsequent to adverse events, such as blood tests, imaging, visits with consultants other than medical oncologists, supplementary biopsies and endoscopies or hospitalization were also included. Costs were divided into seven different categories for analysis, as defined in Table 1. 

Five patients in this cohort were treated in the BMS CA 209,401 clinical trial, a Phase 3b study testing ipilimumab–nivolumab for metastatic melanoma patients in first-line setting. Since the treatment costs were covered by the study sponsor, an estimation was performed. Some patients needed to be hospitalized due to their general condition and received the treatment at the same time. Since the billing in this case is a hospitalization package, an estimation of treatment cycles received was also added. However, this was not the case for radiotherapy costs for a minority of hospitalized patients who received this treatment during their hospitalization. As they are also part of the billing package, separate radiotherapy costs were not taken into account for inpatients.

## 3. Results

### 3.1. Baseline Patient Characteristics

This retrospective study identified 62 patients treated with ipilimumab–nivolumab (Figure 1). Median follow-up was 32 months (18–1173 days). The majority of patients were males (37/62–60%), and 37/62 (60%) of patients had cutaneous melanoma. The cohort also included 13/62 (21%) melanoma of unknown primary (MUP), 4/62 (6%) mucosal and 8/62 (13%) uveal melanoma. For cutaneous and MUP melanoma, 21/62 (34%) were stage M1c and 21/62 (34%) M1d. For patients with brain metastasis, 16/22 (73%) received stereotactic radiosurgery (SRS), either before the treatment start or during follow-up. The vast majority, 50/62 (81%), received the combination as a 1st line treatment and 34/62 (55%) had tumors that were BRAF-V600 positive. Appendix A shows patient characteristics according to the frequency of irAEs (total no. Grade 1–2 and Grade 3–5). The median age of the total cohort at the start of the treatment was 56 years (19–80). A total of 33/62 (53%) had three or more organs involved. A total of 27/62 (43%) received four cycles of ipilimumab–nivolumab. The majority of patients in the cohort (39/62, 63%) received a maintenance treatment.

### 3.2. Clinical Safety

Among our patients, 54/62 (87%) presented at least one irAE, 31/62 (50%) presented a Grade 3–4 irAE and one patient died from an irAE (pneumonitis) following ipilimumab–nivolumab. The most common irAEs were diarrhea with 23/62 (37%) at all grades and 8/62 (13%) at Grade 3–4, hepatitis with 22/62 (35%) at all grades and 9/62 (14%) at Grade 3–4, and skin rash with 21/62 (34%) at all grades and 6/32 (10%) at Grade 3–4. No cardiac or ocular toxicities were reported in this cohort. The irAEs found, as well as their frequencies, are similar to those reported in the literature [9]. Among patients that had an irAE, 11/54 (20%) did not require steroids or other immunosuppressive drugs (ISDs), 36/54 (67%) received systemic steroids and 7/54 (13%) needed an additional ISD. Additional ISDs given were anti-TNF alpha, micophenolate mofetil or intravenous immunoglobulin (IVIG) in one case of polyneuropathy (Guillain–Barré syndrome).

The median treatment-free interval was 127 days (4.2 months), defined by the time between the end of the therapy of interest (last dose) and the first dose of the subsequent systemic therapy. A total of 32/56 (58%) required a subsequent treatment. Most patients (18/32) received a BRAF inhibitor (BRAFi)/MEK inhibitor (MEKi) combination (average duration, 82 days). Eight patients received subsequent immunotherapy (anti-PD1 only, talimogene laherparepvec (T-VEC)), autologous transfer of tumor-infiltrating lymphocytes (TILs), with a longer duration (average duration, 254 days). Four patients received MEKi only (average duration, 103 days) in the uveal melanoma group and two patients received chemotherapy (average duration, 70 days).

The hospitalization rate due to toxicity was 45% (28/62). The mean duration was 10.8 days and the median duration of hospitalization for toxicity was 5.5 days. A total of 27/62 (43%) patients completed the four cycles of the induction phase. A total of 37/62 patients (60%) interrupted treatment due to toxicity and 29/62 (47%) discontinued the treatment due to toxicity in accordance with what has been reported in the literature [9]. A total of 18/62 (29%) of patients discontinued treatment because of progressive disease (PD). Among the causes of hospitalizations related to toxicity, diarrhea was the most frequent (29%), followed by hepatitis (21%) and neurotoxicity (11%).

### 3.3. Efficacy

The overall response rate was 28/62 (45%), with 14/62 (22.5%) complete response (CR) and 14/62 (22.5%) partial response (PR) (Appendix A). As shown in Figure 2, there is an association between toxicity and response rate. The majority of CR patients 12/14 (86%) and 21/28 (75%) of overall responders presented a Grade 3–5 toxicity, and there were no responses in patients without toxicity. This relationship is to be taken with caution, as early PD patients receive relatively fewer cycles and, hence, have little time to develop irAEs. Of note, early PD patients with no toxicity represent a small fraction with only four patients in our cohort. In addition, toxicity does not imply response, as only 28/54 (52%) of patients with toxicity (all grades) and 21/31 (68%) (Grade 3–4) responded (Figure 2).

Progression-free survival (PFS) for patients with Grade 1 or no toxicity was worse than for those with Grade ≥ 2 toxicity (respectively 1.2 and 7.1 months; hazard ratio (HR): 0.22; *p*-value < 0.001; Figure 3A), and remained significant when time-adjusted (2.7 and 7.3 months; HR 0.32; *p*-value = 0.003; Figure 3B). Overall survival (OS) was also worse than for patients with Grade 1 or no toxicity than for patients with Grade ≥ 2 toxicity (6.4 months and not reached; HR 0.27; *p*-value = 0.0039; Figure 3C), closely missing statistical significance when time-adjusted (20.3 months and not reached; HR 0.43; *p*-value = 0.069; Figure 3D). Treatment-free survival (TFS) in the group with Grade 1 or no toxicity was also worse for patients with Grade ≥ 2 toxicity (respectively, 0.8 and 10.7 months; HR 0.15; *p*-value = 7.5 × 10^−6^); and remained significant when time-adjusted (respectively, 0.9 and 12 months; HR 0.15; *p*-value = 9.2 × 10^−6^).

### 3.4. Costs

Our first striking finding is that toxicity costs represent only 3%, on average, of the total expenses per patient, as shown in Figure 4. The most significant contributions were medication costs of ipilimumab–nivolumab treatment (44%) and disease costs (39%, mainly non-toxicity-related hospitalization costs). The lowest per-patient total cost, EUR 1895, was observed in the surveillance category. The highest was the medication cost of ipilimumab–nivolumab treatment, EUR 70,454 per patient. We note the relatively small contribution of the toxicity-related costs (around EUR 4342 per patient) compared to the medication costs of ipilimumab–nivolumab treatment and the disease costs.

Since follow-up times varied between 18 to 1173 days, we also calculated the median costs per week of follow-up and compared to the response and severity of toxicity. Figure 5A shows that CR patients had the lowest global cost per week (EUR 2425/week) despite the associated toxicities. Conversely, PD patients had the highest cost per week (EUR 8325). PR and SD patients had similar intermediate costs (~EUR 3500/week). Thus, the general trend is that good response implies lower weekly costs. Overall, the median toxicity cost per week is again insignificant compared to the two most expensive categories. Disease costs in patients with CR were lower (EUR 1212/week) compared to PD patients (EUR 2979/week) with a similar trend for medication costs (EUR 969/week for CR patients compared to EUR 4724/week for PD patients).

Next, we examined the relationship between maximum toxicity grade and median cost per week (Figure 5B). Interestingly, we observed the highest median costs per week in the absence of toxicity (EUR 9922/week) and a progressive decrease going up to Grade 3 (EUR 2850/week). Grade 4 was associated with a slightly higher weekly cost (EUR 3266/week), and the single patient with Grade 5 toxicity incurred again a high cost (EUR 6043/week). As expected, the toxicity-related cost itself increased with high grades but the trend in total cost per week is instead driven by the disease cost (decreasing from EUR 3855/week with no toxicity to EUR 1408/week at Grade 3) and the medication cost (decreasing from EUR 4932/week in the no toxicity group to EUR 1142/week at Grade 4).

## 4. Discussion

This study analyzes the overall costs of managing irAEs associated with ipilimumab–nivolumab melanoma therapy in a real-world setting in Switzerland. We note that while the price of the combination drugs can vary in other countries, the costs of hospitalizations and RT can be expected to vary approximately proportionally, thus, minimally affecting the reported cost ratios. Regardless, combination therapy always comes with an increased direct medication cost compared to single-agent therapies. In addition, since the combination has a high rate of irAEs, it was expected that managing them would result in a significant rise in treatment costs as well. Surprisingly, the financial impact of toxicities is very small compared to the other costs. This finding has also been reported by Federico Paly et al. [22] in Japan. They estimated irAE costs by combining the rate of Grade 3 events reported in CheckMate 067 with hospitalization rates and drug costs and found that toxicity costs were also a small proportion of total cost. Other groups have addressed toxicity costs in a real-world setting in cost-effectiveness studies [22,23,24,25]. While in our study the irAE cost were extracted with a different, much more detailed method, for all grades we had similar results. Medication costs and disease costs (mainly disease-related hospitalization costs, not toxicity-related) represent the most significant contributions. Of note, costs related to subsequent treatment lines were not included in this analysis, although such data would have provided interesting perspectives on the global treatment cost per patient. Acquiring accurate data on oral therapies such as BRAF and MEK inhibitors is challenging because they are often provided by external pharmacies. Therefore, in this study we have chosen to focus on the ipilimumab–nivolumab combination, since it has the highest cost impact.

Analyzing weekly costs in relation to response to treatment shows that patients with better responses generate lower costs, due, in part, to decreased disease costs, as expected, but to a larger extent due to decreased medication costs. Concurrently, we observe that toxicity severity is anti-correlated with the weekly overall cost. This can be explained by the fact that high-grade toxicities: (i) often imply treatment termination; (ii) are often linked to a better outcome.

As expected for any real-world study, our analysis suffers from a number of biases that need to be examined carefully. First, we list some factors that may have led to toxicity cost underestimation. When a patient had an irAE, billing of expert evaluations was correctly added to the toxicity costs. However, in some cases there was only a phone discussion between physicians, so not all expert opinions were added to the costs (endocrinology), but the estimated cost addition is insignificant compared to other contributions. Medical assessments performed by oncologists after the end of a toxicity (i.e., steroids tapering, liver tests monitoring) could not be included in the toxicity costs but were counted as surveillance costs (if treatment was discontinued) or disease costs (if treatment was ongoing). Topic and oral medications (ambulatory medications) were not included in the cost calculation. Furthermore, patients who had a toxicity that required hospitalization in other hospitals did not have associated costs added to the toxicity cost since the billing information from external hospitals was not available. This was the case for three patients: one with meningitis, one with hepatitis and diarrhea and one with encephalitis. This is also the case for external laboratory and radiological evaluations. However, the majority of patients in our cohort had a complete follow-up at CHUV. Overall, we estimate that the potentially missing cost components could be noticeable in some cases, but could in no way influence the qualitative comparisons to other types of costs, such as the medication cost or the disease cost.

We note also that the disease cost may be overestimated in case patients were followed in our hospital for other comorbidities during their period of follow up. The associated costs were added to the disease cost because of the difficulty to untangle the origins of the costs in the billing system. However, we expect this cost addition to not have qualitative impacts on our results. This study did not attempt to compare with anti-PD1 monotherapy. Finally, this study is limited by its retrospective nature; however, the observed rate of severely delayed irAE and response rates were similar to the literature [9], showing that our population is representative. In a broader perspective, it would be interesting to compare the direct costs to the healthcare system to the potential socio-economic benefits for responders. This is particularly relevant for melanoma patients due to their relatively young age (mean 56 y in this study) and their potential to further contribute to society if they respond well to treatment.

## 5. Conclusions

The cost of toxicities was unexpectedly insignificant compared to the total costs and, in particular, medication costs. In addition, patients with a higher degree of toxicity typically have lower costs and better outcomes. Although this is a small retrospective cohort and the observed results should be validated prospectively, this study evidences clear trends in the costs of managing irAEs of ipilimumab–nivolumab in a real-world setting.

## Figures and Tables

**Figure 1 cancers-15-00031-f001:**
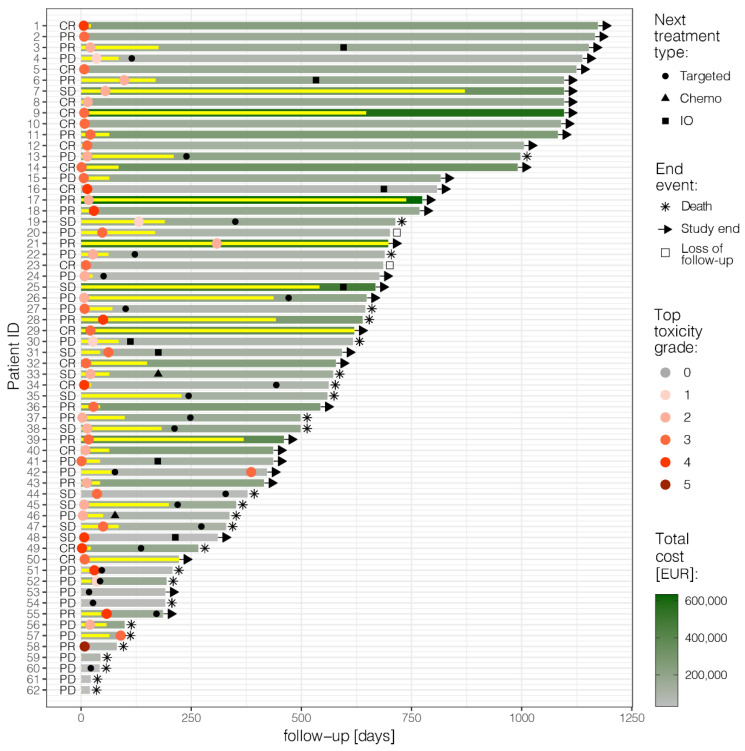
Swimmer plot: follow-up of each patient from the start of ipilimumab–nivolumab until death, study end, or loss of follow-up; treatment duration (yellow bars); best overall response following RECIST 1.1 (CR: complete response, PR: partial response, SD: stable disease, PD: progressive disease); maximum toxicity grade experienced (red dots); next treatment line (black symbols); and total cost (shades of green). Figure produced in R v4.0.3 [19] using ggplot2.

**Figure 2 cancers-15-00031-f002:**
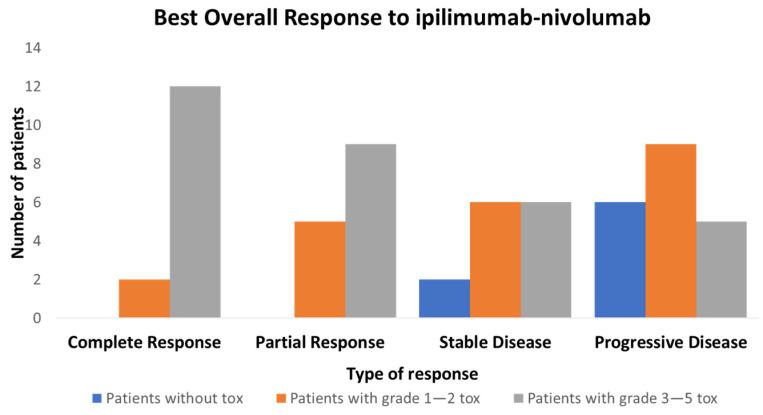
Relationship between best overall response (BOR) following RECIST 1.1 and maximum toxicity grade divided in 3 groups: patients without toxicity, patients with Grade 1–2 toxicity and patients with Grade 3–5 toxicity.

**Figure 3 cancers-15-00031-f003:**
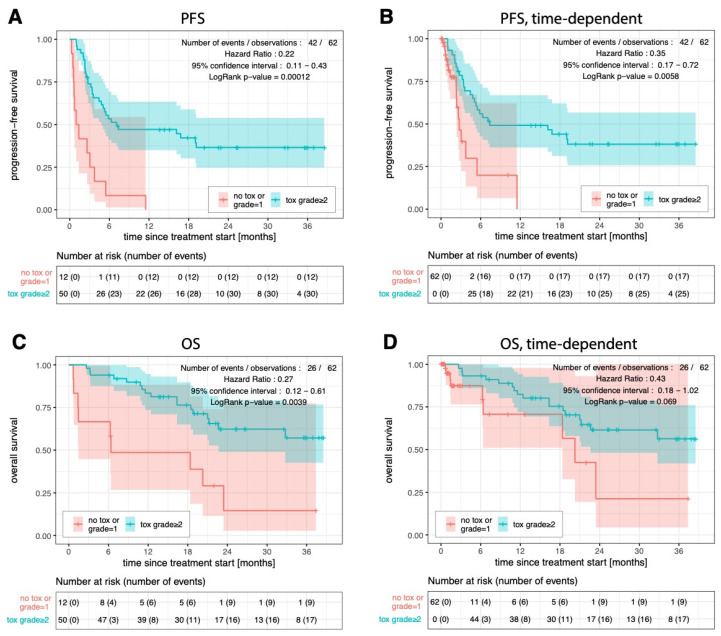
Progression-free survival considering the occurrence of toxicities of Grade ≥ 2 as a fixed covariate (**A**) or as a time-dependent covariate (**B**). Overall survival with fixed (**C**) or time-dependent (**D**) covariate. Kaplan–Meier and Cox regression analyses were performed in R v4.0.3 [19], using the survival and survminer packages. The log-rank test was used to determine statistical significance of survival curve dissimilarity.

**Figure 4 cancers-15-00031-f004:**
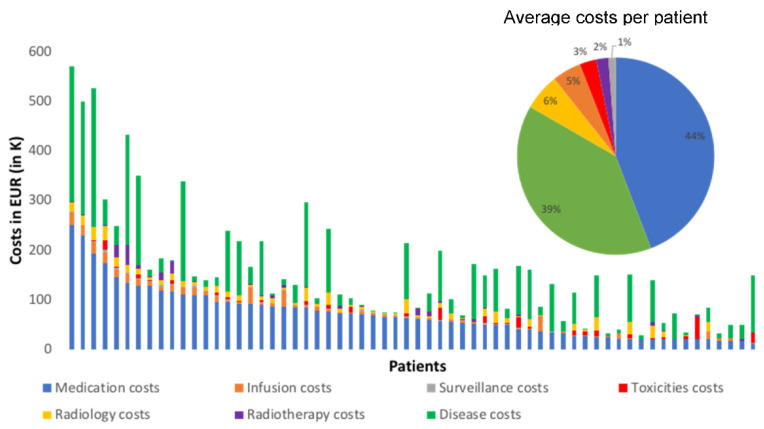
Total cost per patient with contributions of the cost categories defined in Table 1. Inset: average proportion of each cost category over the whole cohort (same color code as main graph).

**Figure 5 cancers-15-00031-f005:**
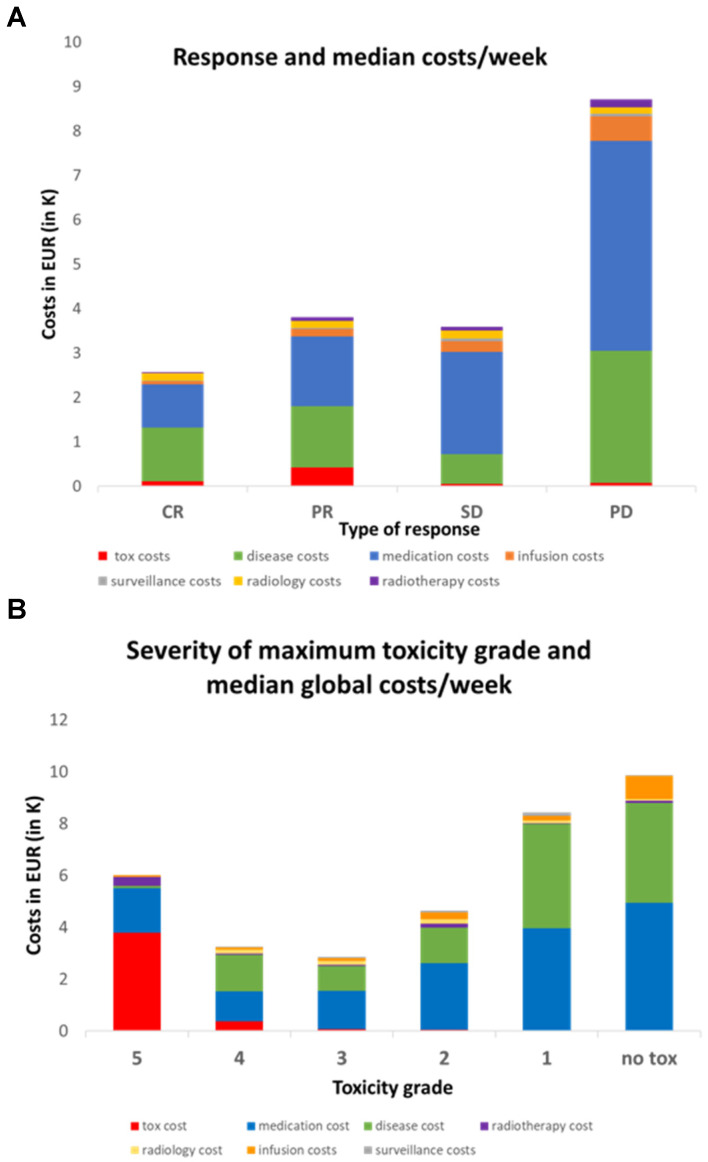
(**A**) Relationship between median cost per week and best overall response following RECIST 1.1. CR: complete response, PR: partial response, SD: stable disease, PD: progressive disease. (**B**) Relationship between median cost per week and maximum toxicity grade.

**Table 1 cancers-15-00031-t001:** Definition of cost categories used to estimate ipilimumab–nivolumab treatment-related expenses.

Total cost	Total billing at the CHUV hospital during the follow-up period. Costs of subsequent treatments were not included.
Medication Costs	Cost of ipilimumab–nivolumab combination +/− nivolumab maintenance drugs for specific patients (doses administered depending on weight).
Infusion costs	Infusion, laboratory, medical and nurse visits related to the infusion day.
Surveillance costs	Medical and nurse visits and laboratory analysis occurring between the drug-infusion days during the concomitant phase. Additional procedures or hospitalizations in between the concomitant phase.
Toxicity-related costs	Global cost of toxicities (hospitalizations related to toxicities, cost of investigational and treatment procedures, expert evaluations).
Radiology cost	Cost of all radiology exams performed during follow-up, except those related to toxicities.
Radiotherapy cost	Cost of radiotherapy treatments.
Disease costs	Clinical and laboratory assessments during the follow-up period; hospitalizations due to disease, ambulatory costs from other experts due to disease, ambulatory biopsies, and were calculated as the difference between total costs and the sum of medication, infusion, surveillance, radiology and radiotherapy costs

## Data Availability

The data managers of the Oncology Department compiled the clinical, financial data and adverse events with a single REDCap code for each patient. Data available upon request provided appropriate approval by the Ethics Committee.

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
