# Peer review of "Estimated Costs of the Ipilimumab–Nivolumab Therapy and Related Adverse Events in Metastatic Melanoma"

_cancers, 2022, doi:10.3390/cancers15010031_

Round 1

Reviewer 1 Report

The paper analyzes the overall costs of managing irAEs associated to ipilimumab- 332 nivolumab melanoma therapy in a real-world setting, in Switzerland. The authors have conceived a scientifically sound  study which emphasized the surprisingly low cost of toxicological reactions versus other costs. The methods were presented in detail and investigated all aspects of the topic, such as clinical safety, efficacy, costs. The authors have also discussed the bias of their study, thus offering other researchers the opportunity to conduct further studies with improved parameters. The paper is fluently written, in a comprehensible yet academic style, which makes it easily readable. 

Therefore, it is my opinion that the paper can be published in its current form.

Author Response

Point 1

Therefore, it is my opinion that the paper can be published in its current form.

Response 1

Thank you!

Reviewer 2 Report

This study by Moura et al, seeks to investigate the actual costs of the immunocombination therapy; ipilimumab-nivolumab. The authors have done extensive work to clarify all expenses concerning this therapy, and they present the data in a clear and well-written manuscript.

General considerations:

I believe that it is important with a focus on the expenses of the modern and very expensive therapies emerging in many areas of oncology these days, wherefore I have read the manuscript with interest. However, I do miss that the overall expenses from this therapy take into account that many of the patients responding to therapy will be able to contribute to society in the future, especially since many of these patients are young and still have many years on the labor market (average age 56 years in this paper). Also, second-line therapies are still very expensive, and I do miss more information on why this has not been included in the paper – or at least a more thorough discussion on this (p.9, l.348-9). 

Minor language corrections:

P.1, l.24: …(irAE) of in a real-world setting.”of” should be deleted

p.7., l. 273 Treatment-free survival (TFS) in the group with grade 1 or no toxicity was also worse than for patients with grade≥2 toxicity. Than should be added

Comments for figures:

Fig. 1: The symbol for death and study end is not the same in the figure as in the text. Also, I do not understand what “study end” covers?

Fig. 5B: In the text, it is noted that the less severe toxicity grades, the higher the costs. Looking at the figure grade 4 (and grade 5) has higher costs than grade 3 indicating that maybe there is a difference on the very severe toxicities (gr. 4-5) and grade 3. This should be discussed or the numbers should be illustrated differently. And this is the same for fig. 5A where there is no (or in favour of SD) difference between patients with PR or SD (p.8, l. 308-315).

Fig. 6: Not interesting, suggest it should be removed

Author Response

  • I believe that it is important with a focus on the expenses of the modern and very expensive therapies emerging in many areas of oncology these days, wherefore I have read the manuscript with interest.

We thank the reviewer for this very positive assessment of our manuscript.

  • However, I do miss that the overall expenses from this therapy take into account that many of the patients responding to therapy will be able to contribute to society in the future, especially since many of these patients are young and still have many years on the labor market (average age 56 years in this paper).

We thank the reviewer for pointing this out. Indeed, melanoma often affects young patients and our cohort is representative of the real world melanoma data. However, the aim of our trial was not to do a cost-effectiveness analysis. We analyze deeply the direct costs to the healthcare system, such as medication costs and toxicity costs, but unfortunately, analyzing further indirect costs or benefits remain out of the scope of this paper. Our results are already very useful to settle arguments we hear sometimes about the ipi-nivo combination being too expensive due to side effect management costs: we now know it is wrong. We agree however that it would be very interesting to see the broader impact of these expensive treatments on society, although this would be a topic for a further cost-effectiveness analysis including socio-economic components. To echo the Reviewer’s comment, we added the following text at the end of the Discussion:

In a broader perspective, it would be interesting to compare the direct costs to the healthcare system to the potential socio-economic benefits for responders. This is particularly relevant for melanoma patients due to their relatively young age (mean 56 y in this study) and their potential to further contribute to society if responding well to treatment.

  • Also, second-line therapies are still very expensive, and I do miss more information on why this has not been included in the paper – or at least a more thorough discussion on this (p.9, l.348-9).

We agree with the Reviewer’s comment. We also initially wanted to analyze second-line therapies to have a global cost of the treatment per patient. However, oral therapies, such as BRAF and MEK inhibitors, are given by external pharmacies in Switzerland, and consequently  it is challenging to obtain their costs. Therefore, in this study, we have chosen to focus on the ipi-nivo combination, since it has the highest cost impact. We added text to the Discussion to address this point:

Of note, costs related to subsequent treatment lines were not included in this analysis, although such data would have provided interesting perspectives on the global treatment cost per patient. Acquiring accurate data on oral therapies such as BRAF and MEK inhibitors is challenging because they are often provided by external pharmacies. Therefore, in this study we have chosen to focus on the ipilimumab-nivolumab combination, since it has the highest cost impact.

4) Minor language corrections:

P.1, l.24: …(irAE) of in a real-world setting.”of” should be deleted

p.7., l. 273 Treatment-free survival (TFS) in the group with grade 1 or no toxicity was also worse than for patients with grade≥2 toxicity. Than should be added

 We thank the reviewer for pointing these errors out. We corrected them.

5) Comments for figures:

Fig. 1: The symbol for death and study end is not the same in the figure as in the text. Also, I do not understand what “study end” covers?

Thank you for this excellent remark. We have now corrected the symbol to be the same in the figure and in the text. “Study end” means the end of the follow-up (August 31st 2019).  

6) Fig. 5B: In the text, it is noted that the less severe toxicity grades, the higher the costs. Looking at the figure grade 4 (and grade 5) has higher costs than grade 3 indicating that maybe there is a difference on the very severe toxicities (gr. 4-5) and grade 3. This should be discussed or the numbers should be illustrated differently. And this is the same for fig. 5A where there is no (or in favour of SD) difference between patients with PR or SD (p.8, l. 308-315).

Thank you for your excellent suggestions. We rewrote the description of Fig 5B, which was indeed not optimal. It now reads:

Interestingly, we observed the highest median costs per week in the absence of toxicity (€9,922/week) and a progressive decrease going up to grade 3 (€2,850/week). Grade 4 was associated with a slightly higher weekly cost (€3,266/week), and the single patient with grade 5 toxicity incurred again a high cost (€6,043/week). As expected, the toxicity-related cost itself increased with high grades, but the trend in total cost per week is instead driven by the disease cost (decreasing from €3,855/week with no toxicity to €1,408/week at grade 3) and the medication cost (decreasing from €4,932/week in the no toxicity group to €1,142/week at grade 4).

Regarding Fig. 5A, we think there is no significant difference between SD and PR, so we now describe them collectively as having an intermediate value. To clarify, we added the text:

PR and SD patients had similar intermediate costs (~3,500/week). Thus, the general trend is that good response implies lower weekly costs.

7) Fig. 6: Not interesting, suggest it should be removed

As suggested by the reviewer, we have removed Figure 7.
